# Protective Support and Supportive Protection: Critical Reflections on Safe Practice and Safety in Supervision

**Carmel Devaney** *[ID] and **Caroline Mc Gregor** [ID]

UNESCO Child and Family Research Centre, National University of Ireland, H91 TK33 Galway, Ireland; caroline.mcgregor@nuigalway.ie
* Correspondence: carmel.devaney@nuigalway.ie

**Abstract:** This paper was based on a framework for practice and supervision based on 'protective-support and supportive-protection' (PS-SP) that can be used to discuss and plan for practice in a way that maximises the capacity of workers in child protection and welfare (CPW) services to provide support and protection simultaneously. The framework is underpinned by a long-established assumption about social work in child protection and welfare as a socio-legal practice of mediation in the social. The PS-SP framework was initially developed within an ecological context with a focus on networks and networking. In this paper, we developed this framework further, framing practice supervision using four functions of supervision (management, support, development, and mediation) and including a fifth component on safety. We did this alongside a review of related considerations around safety in supervision in general and in child protection and welfare in particular. While noting the important contribution of this work, we identified ongoing gaps for supervision focused on safety when considered within an ecological context. Bearing in mind the well-evidenced stress, challenges, and vicarious nature of child protection and welfare practice, we argued the importance of a wider framework based on PS-SP for supervision and support to manage this complexity with a particular emphasis on 'safety' as a contribution to this. To illustrate our framework and discussion, we referred to a case study throughout. This case study is drawn from a high-profile child abuse inquiry in Ireland over a decade ago. This case was chosen as it demonstrates the complex interplay of needs for support and protection over extended time. We considered how the PS-SP framework may be used in the present (hypothetically) regarding such a case scenario.

**Keywords:** supportive protection; protective support; supervision; safety; practitioner; child protection; welfare; support

## 1. Introduction

As Spratt et al. (2015) contend "child protection is the public issue of our time with increased public scrutiny of practice driven by public service enquiries and changing political and economic contexts" (citing Fernandez 2014, p. 801; Heino 2012, p. 157). Despite being such a well-researched area globally, child abuse and neglect, or child maltreatment remains a major social problem that requires interventions at individual, family, community, and societal levels. There are many and ongoing critiques of practice in this area, often in response to high-profile tragedies that diminish public and media confidence in the capacity of systems and professionals to respond appropriately to child abuse and neglect. A major challenge for systems and for practitioners is balancing the delivery of family support and child protection services with diversity in the orientations of child welfare systems evident globally (Gilbert et al. 2011; Merkel-Holguin et al. 2018; Canavan et al. 2022). One may argue that this has been advanced with the greater expansion of use of well-established models of child protection practice, such as Signs of Safety (Turnell and Murphy 2017) or Reclaiming Social Work (Cross et al. 2010) and family support practice, such as the Meitheal model in Ireland (DCEDIY 2022) that emphasises a strengths and partnership approach.

However, it can also be argued these models and associated supervision practices tend to focus on the micro-level of practice, without considering the impact of wider social factors. This is despite it being well-established that child welfare services aimed at support and protection need to be reoriented firmly towards an inequalities framework in light of the overarching evidence from studies such as Bywaters and the Child Welfare Inequalities Project Team (2020). For example, in Signs of Safety (SoS), the main role of the supervisor is to ensure fidelity and support the implementation of the practice model for families where it seems there is little or no focus on safety from the point of view of the practitioner (Keddell 2014). In our framework, use of the ecological model, within a networking context, is offered as one way to address this. Another limitation is that when applying different models of practice, most often, they do not fully address the inter-relationship between support and protection and can tend to be used in isolation or separately as a 'family support' or a 'child protection' model. For example, while Meitheal and Signs of Safety have many common features in the Irish context (Malone et al. 2018), there is no formal attempt to consider their interrelationship in the current model for integrated services. In the development of our model, as explained below, we attempted to contribute to informing how we can better integrate our approaches to protection and support.

We begin, in Devaney and McGregor (2017), with an historical overview of the history of family support (Devaney 2011) and history of child protection (Skehill 2007) to highlight the intrinsic interconnection between the histories of what we commonly refer to as 'family support' and 'child protection' and the nature of practices therein. Following on from this, in (McGregor and Devaney 2020a), we used a range of data sources to demonstrate that many families are 'in the middle' with regard for needs for support and protection. We followed this with an article (McGregor and Devaney 2020b) that broadened the consideration to how a practice and supervision framework based on protective support and supportive protection can incorporate the ecological model and a networking approach to take account of these wider factors more effectively in one-to-one practice and supervision.

From this work, the PS-SP framework of practice and supervision has emerged. The PS and SP framework advocates that children and young people are best protected by providing responsive support to them, to their parents, their family, and their wider networks. It further holds that practitioners working in CPW systems have both the responsibility and capacity to provide protective supports and supportive protection to children, young people, parents, and families. This framework is based on the fact that most families involved with the CPW system have changing levels of need and risk over time and are simultaneously in need of both support and protection (Devaney et al. 2021). As families' circumstances and levels of risk and need change over time, so too must the system's response to this. Families who receive the appropriate support response in a timely manner can progress to managing their challenges, functioning well, and ensuring children are protected and thriving. However, situations can change, relationships can deteriorate, and levels of need and risk can rise again. In order to emphasise the fluidity needed in a CPW system which reflects the need for 'protective support and supportive protection' and to respond to the complexities and changing levels of need and risk in families' lives over time, the authors adapted the four-tiered Hardiker model (Hardiker et al. 1991) to reflect this dynamic, open continuum, and to also include and account for the important role of informal social support in helping family members. The framework adds value in three key areas: it usefully focuses attention on the majority of 'families in the middle' and their fluctuating need for supportive as well as protective responses (categorised as Level 2–4a) and it brings 'informal support networks' (with this new Level 1a) into the frame of analysis, acknowledging the role of family, wider kin, and community in preventative support. It also crystallises that while 'universal' or 'Level 1' services are mostly supportive, there is also a need for a sharper emphasis on needs for protection rather than seeing this as something to be considered when the family's level of need is classified as Level 3 or 4 only. It also offers a way to shift beyond the negative impact of an over-focus on risk, especially in terms of how threatening this can be to families, by emphasising that

even at Level 3 and 4, where the focus is on legal protection, extensive supportive work remains essential. The framework is a scaffold for analysis, reflection, and thinking about a situation that seeks to address the known limits of support services that tend to refer to rather than engage with risk issues in a way that minimises emphasis on the supportive relationship and need for an integrated approach. Thirdly, it offers a framework to inform practice by highlighting that practitioners who are offering mostly support services need to be confident and competent in addressing child welfare concerns (beyond just legal requirements or mandatory reporting). Likewise, those offering risk management and protection services need to have excellent skills in supportive, relationship-based practice.

However, while providing an in-depth guide to how one might map and begin to address these issues, we suggest the framework, as it stands, remains too broad. To focus the work more stringently, we paid more attention in this article to the processes of supervision using Dr Tony Morrison's four functions of supervision, namely management, support, development, and mediation (Morrison and Hathaway 2005). We also argued that a greater focus on 'safety' as a core defining feature of supervision is needed based on the work of McPherson et al. (2016) situated within a wider ecological and network-based context. We emphasise that PS-SP is not another practice model. It is a broad framework for critical analysis, reflection, and thinking about the complex field of supporting and protecting children and families that can scaffold practice and supervision discussions.

With this starting point in mind, this paper considered how the broad framework of protective-support and supportive-protection in child protection and welfare (Devaney and McGregor 2017; McGregor and Devaney 2020a, 2020b) can complement and support the current demands and responsibilities of child welfare practitioners that pays attention to safety as a defining focus. We did this in the following ways.

In Section 2, we update and review international literature on child welfare orientations and practices that demonstrate a continued focus on 'interface' or 'integration' of support and protection. In this, we include a commentary on how safety and safe practice are conceptualised. We suggest our existing PS-SP framework contributes to this by providing a way to reflect on protection and support in an integrated and dualistic matter. In Section 3, in order to inform this development, we review current literature on supervision in general and safety in supervision in particular. We underline the contribution of this work but also highlight the limits of a focus on psychological or therapeutic 'safety'. In Section 4, our discussion, we consider the knowledge base to date on PS-SP and on supervision and safety and consider how we can advance our thinking and practice based on this. To illustrate our framework and discussion, we refer to a case study throughout. This case study is drawn from a high-profile child abuse inquiry in Ireland. This case is chosen as it demonstrates the complex interplay of needs for support and protection over extended time. We consider how the PS-SP framework may be used in the present (hypothetically) regarding such a case scenario to inform the application of the framework to practice scenarios. We propose ways in which the PS-SP framework can complement existing supervision models, extended to include a wider ecological and networked context. We conclude that no matter how many new frameworks we debate and advocate, it is down to the practitioners and those who supervise and support them to implement it in very complex contexts and how they have to feel safe in doing so.

## 2. Orientations in Child Welfare: Implications for Practice

As discussed in previous work and updated here, there is an array of literature to inform us on the orientations of child welfare systems and which considers how child protection and support is organised and delivered. Below we discuss some of the prominent themes in this literature, highlighting the complexities involved and raise the question of how practitioners operating in these systems and doing the direct work are supported and supervised in relation to safe practice that takes account of this complex context. We suggest that the PS-SP framework offers a useful tool, which resonates across jurisdictions

and orientations, to help mediate and manage child protection and welfare practices and systems.

It is well known that child protection and welfare systems are shaped by their national and regional context. Today, in nearly all jurisdictions, there is a marked emphasis on rights (Lonne et al. 2021; Spratt et al. 2015; Gilbert et al. 2011). This is true for established systems such as the Irish welfare and protection system (Burns and McGregor 2019; Tusla 2019), the devolved regions of the United Kingdom (Bunting et al. 2017, p. 6), the Netherlands (López et al. 2018), France (Bolter and Séraphin 2019), and evolving systems such as those in Hungary and Romania (Anghel et al. 2013, p. 248), Mexico (Valencia Corral et al. 2020, p. 3), and India (Rotabi et al. 2019). Of relevance to this paper is the emphasis on a right to both support and protection as needed.

Gilbert et al. (2011) and Collins (2018, p. 371) have noted the "pendulum swing" between prevention and protection throughout systems and identify public agenda setting as a driver in the orientation of systems. The way support and protection are implemented has been advanced with current developments in research relating to intervention, prevention, and public health models. For example, Malone and Canavan (2021) noted that the current trend in practice is orientated towards intervention and prevention, while Lonne et al. (2021, p. 12) pointed out that a public health paradigm of universal prevention approaches is emerging. This universal preventative approach is held by some to be an effective response to the prevalence of child maltreatment and systemic failures (Daro 2019, p. 17; Herrenkohl et al. 2020; Churchill and Fawcett 2016). There are ongoing challenges, however, in the idea of replacing one system type with another, with greater efforts required to support practitioners and service managers to examine how best to mediate and manage practice in the context of an ongoing and dynamic interplay between support and protection. In Ireland, for example, while there has been a shift to greater attention to support, prevention, and early intervention since the establishment of Tusla -Child and Family Agency (Tusla), like many other child welfare systems, the focus remains within a narrower child protection focus (Canavan et al. 2022). The implementation of the Meitheal model as a national practice model for family support (DCEDIY 2022) and Signs of Safety (Turnell and Murphy 2017) as a national model for child protection within Tusla has reinforced a positioning of 'support' and 'protection' both conceptually and in practice as 'interfacing' rather than 'integrated'. This is despite both models deriving from very similar core values and principles (Malone et al. 2018). While helpfully differentiating between levels of need and associated service response, the day-to-day reality for practice and practitioners is that the majority of child welfare interventions are with 'families in the middle' who are usually in need of both support and protection either at the same time or at different times, moving between levels and thresholds (McGregor and Devaney 2020a) with children and young people at risk of abuse or neglect protected by timely, appropriate support and protection. However, the complexity of addressing the need for support and protection across a person's or family's whole eco-system requires a more structured and detailed framework than is presently available, as discussed in (McGregor and Devaney 2020b). With our framework, our argument is that it should not be about 'either/or' as if the systems re-interfacing and separate, but rather that we place PS-SP as an overarching framework to better inform practice using the well-defined thresholds and continuums of practice that are essential for decision making, needs analysis, and service delivery. In McGregor and Devaney (2020a), we argued that the Hardiker model (for example) should be developed to incorporate differentiation of high-risk work that comes into the PS-SP frame and which aligns more within a criminal justice model. We also suggested the addition of the role of informal support and protection referred to later. Here, we illustrated how the model can be applied specifically in relation to safety and safe practice in supervision and practice development (p. 28).

The facts are that despite in-depth analysis of many dimensions of child welfare practice, there continues to be widespread evidence of increasing rates of referrals to child protection and welfare systems in the international literature due to wider definitions of need

and risk, which broadens the scope for service provision (Canavan and Furey 2019). This has resulted in increased demand on systems, which has led to discussions around differentiation within services in order to provide a service that is responsive, relevant, and appropriate to family needs and circumstances (Churchill and Fawcett 2016; Gilbert et al. 2011). This can be seen in dualistic approaches that are tailored to the particular needs of cohorts who are either most at risk with an acute need for protection, or a those with non-urgent need for welfare interventions (Trocmé et al. 2014, p. 484; Churchill and Fawcett 2016, p. 310). This approach can be seen in the US, Australia, Canada, and Ireland (Spratt 2008, p. 420; Trocmé et al. 2014, p. 484; Malone and Canavan 2021).

Many authors have reflected on the effectiveness of these various orientations towards service provision. Kojan and Lonne (2012) compared the social democratic Norwegian child protection and welfare system to the neoliberal Australian system, which is orientated towards protection from risk of harm and neglect in order to consider the effectiveness of their respective regimes. They note that the Australian system emphasises procedures and standardised assessment with increasing investigations and risk-averse interventions, resulting in rising numbers of children in out-of-home care. Conversely, even though the significant professional discretion in Norway gives room for contextual solutions, high numbers of children enter out-of-home care. Nonetheless, the authors maintained that as both systems are functioning well there may be room to consider the learning in each one (Kojan and Lonne 2012, p. 105). Pösö et al. (2014) argued that a universal family service orientation may ignore diversity of needs and rights of children needing protection and support. However, in such deliberations, less emphasis is placed on the question of how the practitioner is supported in doing this complex work, whatever the 'orientation' of the system, that demands complex and delicate mediation between support and protection for children, young people, and families. This becomes more problematic when one considers this practice of support and protection within an ecological context.

For example, with regard to culture, which imbues all aspects of the ecological levels, as Rotabi et al. (2019) argued, practice must be grounded in a culturally relevant values base to be effective (see also, Jabeen 2016; Chung et al. 2022). Authors have also called for the integration of family and community-based efforts to support child protection to support a sustainable practice in the safeguarding in the absence of formalised regulated systems in the African literature (Canavera et al. 2016, p. 366). This reflection on cultural strengths in developing systems, including the role of communities, especially indigenous communities and informal networks of support, offers alternative routes to child safeguarding that are culturally acceptable and sustainable (Canavera et al. 2016; Connolly and Katz 2019).

Additionally, thinking of exo- and macro- levels of the ecological system in particular, numerous authors have criticised the individualisation of social problems under neo-liberal system reforms and innovation as a solution to child maltreatment in society. This is because these orientations towards prevention and intervention do not address the structural roots of issues that drive child maltreatment in society due to a focus on issues of family functioning at the expense of material improvements in family lives and communities (O'Leary and Lyons 2021, p. 2; Churchill and Fawcett 2016, p. 315; Bolter and Séraphin 2019, p. 76). The need to rethink child protection and welfare in the context of the reality of the impact of inequality has been well established (Bywaters and the Child Welfare Inequalities Project Team 2020; Maguire-Jack and Katz 2022). This reinforces the importance of an ecological and network frame (McGregor and Devaney 2020b) that recognises the relationship between social context and inequalities and child protection and welfare practice (Bywaters and the Child Welfare Inequalities Project Team 2020; see also El Husseiny et al. 2021; Schmid 2007). It is of particular interest to note that all the discussions above, in one way or another, are concerned with the inter-play between support and protection and this manifests itself in many ways internationally. For example, Katz and Hetherington (2006, p. 429) considered the range of differences between dualistic and holistic systems in the European context. Dualistic systems are child protection focused, and family support is in the main dealt with separately, while holistic systems promote

family support and prevention, on a continuum of care. Ireland provides an example of a holistic system in a traditionally rudimentary regime, as practice in principle often combines child-focused protection and interventions, and family support through a mixed provision of service led by a state agency, combined with provision from community and third-sector organisations (Burns and McGregor 2019, p. 132; Malone and Canavan 2021, p. 6). It is within an Irish context specifically that McGregor and Devaney advocated for further advancement of a 'holistic' approach by framing practice in terms of protective-support (when mostly working in a supportive context) and supportive-protection (when mostly working in a protection context). The underpinning message, as reinforced in international literature, is that most intervention is with the majority of families ('families in the middle') who require both support and protection simultaneously. It is for this reason that we think that the PS-SP framework situated within an ecological context is well placed to enhance both supervision and practice in this area.

Throughout the discussions on orientations in child welfare in general, while much emphasis is placed on practice guidance and requirements, less attention is paid to the support and supervision for the practitioner to deliver safely on this complex issue that is, beyond doubt, a major political and public concern internationally. Overall, we can conclude from the literature that while there is a prevalence of ongoing dynamic change processes across various jurisdictions with an emphasis upon improved professional practice, system integration, and better outcomes for children and families (Lonne et al. 2021, p. 3), there is limited attention paid to the supervision and support for the practitioner to implement these practices and systems on a day-to-day basis. Implicit within the discussions above are concerns about safety practices, policy and practice guidance, safeguarding approaches, and practice models focused on safety (e.g., Signs of Safety). From this synthesis of some of the literature relating to orientations in child welfare and the emphasis on safety therein, it is clear that the practitioner and supervisor operate in a tricky space and have a great deal to consider. The PS-SP framework serves as a way of considering the duality that is referred to in the literature. It promotes an approach that is well beyond interface—as we know that is too simplistic for most realities of child welfare practice—and towards an integrative and interactive approach. Underpinned by children's right to protection, development, support, and survival, the PS-SP framework offers substance and foundation to build on what is already available. As discussed below in Section 3, supervision plays a key role in child protection and welfare practice. However, in these approaches, there is notably limited attention paid to safety when it comes to supporting the people who are charged to deliver on the ground in relation to support and protection. Even where this is present, most notably influenced by the work of McPherson et al., etc., it tends to focus on the micro aspects of safety, for example, relational and psychological aspects. In the discussion, we consider how an emphasis on protective support-supportive protection, with a particular focus on safety across the ecological system, can add to existing models and practices to help address the widely acknowledged public and political problem of improving child welfare outcomes. We acknowledge safety is just one of many core concerns in child welfare intervention and we suggest this framework can be adapted to include a range of 'scaffolds' for critical reflection, analysis, and thinking about how best to intervene, especially in complex cases (thinking specifically about risk management) affected by both individual and socio-structural factors.

## 3. Supervision in Child Protection and Welfare

It is generally accepted that working with children and families is complex, with each set of circumstances and family being unique. Policy makers, managers, practitioners, and academics generally agree that good supervision is essential for high-quality practice in children and family services (Beddoe et al. 2014) and it is recognised as an essential resource for practitioners if they are to provide services which benefit children and their families. In child protection and welfare social work services, supervision is identified as a core feature of practice and is described as a "core mechanism for helping social workers reflect on the

understanding they are forming of the family . . . their emotional response and whether this is adversely affecting their reasoning, and for making decisions" (Munro 2009). As Munro (2011) cautioned, helping families can "never be simply a case of taking an intervention off a shelf and applying it to a family" (p. 44). Expanding on this viewpoint, Thompson (2009) described how "the field of practice is not a static, passive recipient of expert knowledge. The situation itself 'talks back', resists and constrains the practitioner's every move" (p. 319). There is a need, therefore, for high-quality support and supervision to counteract some of the tension and complexity in CPW and to support the work towards meeting the needs of children and families. Munro (2001) described supervision as a core mechanism for critical reflection on the understanding of the family, for workers to consider their emotional response and whether it is adversely affecting their reasoning, and for making decisions about how best to help (p. 53).

Supervision for practitioners in children and family services is typically described as a process which is multidisciplinary, collaborative, and relevant across a range of helping professions (Davys and Beddoe 2010). The goal of supervision is to enhance supervisees' professional knowledge, practice skills, and social functioning, and to develop the quality of professional service that is provided to clients (Bogo and Sewell 2018). Through supervision, social workers' abilities are strengthened, and it is ensured that social workers are held accountable for the services they render. Linking supervision to job satisfaction and the retention of social workers, Carpenter et al. (2013) emphasised that if supervisees have a positive experience of supervision, they are more likely to be motivated and view their role and work within the organisation more favourably. Furthermore, Beddoe et al. (2014) identified supervision as central to good practice and found that supervision contributes to competent professional practices that benefit service users. Moreover, there is also a strong correlation between effective supervision and outcomes for service users with evidence suggesting that supervision may promote empowerment, fewer complaints, and more positive feedback (Akesson and Canavera 2017). This means that service users also benefit in the supervision process because social workers are empowered and encouraged to perform their duties properly.

The literature originally referred to the functions of supervision as administrative, educative, and supportive (Kadushin 1976), while Morrison developed these ideas further and referred to competent, accountable performance or practice (managerial); continuing professional development (developmental); personal support (supportive); and engaging the individual with the organisation (mediation function). Richards et al. (1990) initially added this fourth function—mediation—describing it as the "capacity to act as a representative for the team and to enable others to participate in service delivery" (Richards et al. 1990, p. 14). The meditative aspect of supervision explicitly recognises the complex and competing personal, organisational, and professional agendas present in the supervision encounter (Morrison and Hathaway 2005). Aligned with this point are critiques of the supervision process which note that supervision is used as an opportunity to shape the practitioner into organisationally preferred ways of practice and that it can be solely focused on risk and used as a mechanism to prevent mistakes (Beddoe 2010). Concerns have been raised on the focus of supervision questioning as to whether it is primarily related to performance management. Baginsky et al. (2010) found that local authority managers consider supervision a mechanism for performance management, and although this may result in regular supervision, it makes it less likely the focus will be on support and learning (p. 1280). This challenge is also exacerbated by the complex recording requirements for both practice and supervision (Wilkins 2017). Undoubtedly, the challenging environment of child protection and welfare work necessitates a formal process for workers to review, reflect, evaluate, and plan, with a dual focus of ensuring good outcomes for the child, young person, and family while also attending to the professional and personal needs of the practitioner and the agency or system requirements. Indeed, O'Donoghue (2015) provided the evidence for practitioners' preference for supervision that focuses on their education, support, and practice rather than administrative matters.

Referring to the need for high standards of professional practice, Higham (2006, p. 201) highlighted the multiple social work roles (e.g., planner, assessor, evaluator, supporter, advocate, protector, and manager) that balance empowerment and emancipation with protection and support. Despite its challenges, supervision is welcomed by practitioners and used as a resource to support them in this day-to-day work. The key objectives of supervision are to improve the supervisee's capacity to do the job effectively through educational supervision; to provide clear guidelines on and familiarity with the organisation's mission policies to enable workers to perform their duties; and, lastly, to provide emotional and moral support to the supervisees so that they can have a sense of belonging and satisfaction in their job (Kadushin and Harkness 2014). As noted, practitioners working in child protection and welfare require a safe space for supervision and support to manage this complex and challenging role. In order to protect and support children, young people and families' practitioners must feel, and perceive themselves to be, protected (safe), and supported.

McPherson and colleagues (McPherson et al. 2016) usefully called for organisations to plan for, resource, and prioritise supervision in child and family practice and a conceptual framework that addresses multidimensional nature of practice and supervision. To this end, we include a focus specifically on safety, alongside Morison's four functions of supervision in the context of support and protection. This work advances thinking about safety in practice beyond 'health and safety', keeping children and families safe from a child protection perspective (as referenced above) and physical safety of practitioners in relation to the day-to-day practice. This illustrates how the framework can be used and how other concepts, such as tackling child welfare inequality or working across the life course with families and children can be adapted from this.

## 4. Discussion: Protecting and Supporting Child Protection and Welfare Practitioners through Safe Supervision

Regardless of the orientation or jurisdiction, the evidence is compelling that child welfare interventions and practice is underpinned by a balancing of support and protection within a rights-based framework. We have already discussed the many theories and practice models that exist to conceptualise and to some extent guide this mediation work. However, as Keddell (2014) argued in relation to one of the most well-known models, Signs of Safety, while it has provided practice guidance and supervisory contexts to focus on safety, partnership, and collaboration with families, it remains focused mostly on the micro-level. There is a need for a more expansive framework, within which specific models and approaches can continue to be used, that captures the complex inter-play of protection and support by using an ecological and networked approach as the norm, which is the case in the PS-SP framework that we illustrate in this discussion by reference to a case study example.

### A Case Study to apply the PS-SP framework

The paper now presents key points from an Irish case inquiry report, the Roscommon Child Care Case, in order to consider the application of the PS-SP model in practice and to reflect on its strengths and limitations. This inquiry report (Gibbons 2010) concerned a family of six children whose two parents were convicted of incest, neglect, ill-treatment (Mrs. A), and rape and sexual assault (Mr. A). A key concern in the inquiry report was the extent to which too much emphasis was placed on an optimistic focus on support to the detriment of a sufficient amount of protection for the children (p. 69). A key feature of the case was that there was extensive family support and child protection interventions with the family over many years.

Key points of relevance to this paper from the Inquiry report include:

- The Inquiry Team concluded that the six children were neglected and emotionally abused by their parents until their removal from the home (p. 94). These children were denied their most basic needs for security, food, warmth, clothing, and the loving

care of their parents. They were abused by their parents in their home where they had every right to feel safe (p. 4)

- There is no evidence that either parent understood or sought to consistently meet their children's needs. Both parents, but particularly Mr. A, successfully resisted the efforts of professionals to work in a meaningful way with the children, while appearing to be cooperative on the surface (p. 94).
- Other significant issues raised were an uncertainty that the children were being adequately fed, concern that both parents were drinking to excess, and that the family money was being spent on alcohol rather than food (p. 22).
- There was a belief that these parents could, with support, meet the needs of their children (p. 94).
- The services put in to support the family, although very well intentioned, failed on many occasions to respond fully to the chaos of their daily lives, failed to recognise the risk indicators that arose, and, as a consequence, failed to respond appropriately to the needs of the children (p. 4).
- The six children at the centre of this case were denied their voices on many occasions (p. 5).
- The personnel involved with this family relied on a perceived strong attachment of the children to their parents. They did not recognise classic indicators of insecure disorganised attachment (p. 89).
- A consistent aspect of this case was the attempts by relatives and neighbours to highlight the plight of these children. The concerns expressed by neighbours and family members were consistent with each other and over time (p. 88).

The Inquiry Report also noted:

- Child welfare and protection work is challenging. Child welfare and protection work carries risk. It is not easy to get it right and no person or system will get the balance right all of the time. Most of the services involved with the A family were hopeful that there could be change. That hope is essential to the delivery of services to families experiencing difficulties. However, hope needs to be informed by some evidence of change and of life getting better for children (p. 5).
- Workers should be mindful of the need to consider alternative plans where the desired outcomes are not achieved. In all situations it is important that the case file records the reflective thinking, planning and consideration of outcomes that is guiding the work for the child and family (p. 88).

This case inquiry occurred over a decade ago and the practice issues relate mostly to the late 20th and into the early 21st century. Much has changed since then but the case itself serves as a good illustration of the complex interplay of support and protection needs. We refer to the case as one that can be accessed for learning and reflection purposes and reflect on how an SP-PS approach to practice and supervision may assist in the mindset, context, and knowledge of today.[1]

The late Dr Morrison contended that "the quality of child protection and welfare work will never improve unless agencies understand and invest in high quality supervision" (Morrison and Hathaway 2005, p. 138). In addition, and in order to provide this level of reflective supervision, managers tasked with providing such support require training and knowledge which provides them with the skills to support individual workers (Munro 2011). As Wilkins et al. (2017) noted, to create good social work practice, we need to ensure that social workers are provided with the right supervision, and to ensure this, we have to provide support for managers and create the right systemic conditions. Finding a model of social work supervision that avoids a dialectic between either being a therapeutic, introspective activity or as a tool for surveillance is critical (Manthorpe et al. 2015, p. 3; Beddoe et al. 2014, p. 1). Connecting this to our case example, the complexity of the case would be such that time and space for reflection would need to be structured in a planned way under each of the headings of Management, Mediation, Development and Support with the overarching theme of safety. The PS-SP framework could be used here as a tool for

case management to highlight the 'duality of support and protection' that was needed in this case. This could mean that rather than addressing tasks separately (e.g., a supervision order application, a home help service referral, referral to speech and language etc.) the PS-SP framework could be used as a way to consider the holistic range of services and to plan how the practitioner could network within the ecosystem of this family (which included many actors and services at the meso and exo level) so that a more integrated approach between all services could be developed. A key criticism in the inquiry was the fact that despite the involvement of so many services with the family, there was inadequate coordination and sometimes contradictions that prevented an integrated approach. Even with more well-established tools for intervention in the present day, these issues persist within our systems. For the supervision context, the task would be to shift from thinking about the list of support services and list of risk management/legal tasks and instead to come to the discussion with SP (i.e., ensuring home help is confident raising concerns from observations with the child welfare worker, which do not come within mandatory reporting but do contribute to supporting in a protective way). Likewise, ensuring the solicitors and legal team are fully aware, and trained if needed, in the range of support services available to ensure their protective/legal work does not delink completely from the supportive relational practice. The task for the person in supervision is to reflect on how they can develop their overall approach to see, from the outset, the wider PS-SP framework. This would mean that in using SOS, for example, reflection on how to broaden the concept of safety planning to the wider exo and macro system might be incorporated. In Appendix A, for example, we suggest that Mediation and Support need to be seen as a dynamic process across the eco-system rather than focusing on just one or another level of the systems.

Morrison's model (Morrison and Hathaway 2005) is widely known as the 4 × 4 × 4 model, referring to the four stakeholders in supervision (family members who use the service, practitioners, the organisation itself, and partners' organisations); the four functions of supervision (management, support, development, and mediation); and the four elements of the supervisory cycle (experience, reflection, analysis, action planning). This model acknowledges the interdependence of all four functions of supervision, their impact on key stakeholders, and the four stages of the supervision cycle. Much of the literature on supervision also talks about the importance of a safe relationship between the supervisor and supervisee. The helping alliance which is forged is critical in the change process, as this is where the work takes place, and where change can be attempted (Sanders and Munford 2006). Including safety (McPherson et al. 2016) as the 'fifth' dimension of this well-established supervision model helps in ensuring this safe space for relationships to develop. This safe space allows practitioners to openly consider and reflect on all aspects of their role, the responsibilities and remit of the agency, the circumstances in which the children, young people, and families are living in, the implications of this, the risks and protective factors, and the emotions associated with all of these issues. Indeed, the need for safety has a much wider and, in many ways, more tangible reach. In this paper, we reflected on how the PS-SP model can contribute to this using concrete tools to apply and evaluate the use of a PS-SP approach with an overarching emphasis on safety within supervision frameworks.

For example, using the ecological model as outlined in Appendix A, a focus in the micro- meso sphere could be on maximising supportive protection amongst extended family members. There was involvement of the maternal grandmother, an aunt, and other relatives in the Roscommon Child Care case. A more focused engagement with the family system to maximise support and protection could have been applied. Other considerations at the micro-level would be in the situation where Signs of Safety was being used with the child and the family. By using the PS–SP framework, the concept of 'safety' could be broadened to explore how this could be maximised within the eco-system of the family. A safety plan could be reviewed and revised on the basis of supervision based on the following questions: how does this plan maximise protection in relation to the agreed

supports to this family? How can a supportive approach be maximised with parents while engaging in risk management and protection work with the children?

Reflecting on applying the framework at the exo level, one focus within Management could be about how to have a more consistency across models such as for example Meitheal and Signs of Safety. This could be addressed in practitioner development by initiating greater cooperation or joint training between family support, social work, public health, and other professionals involved.

In relation to mediation, the importance of networking, for example, could be considered in the supervision process. In this case, services were often focused on either the children (child protection and welfare) or the parents (home help) and a networked approach could have brought services together better to offer better support while protecting (e.g., earlier intervention in relation to clear evidence of alcohol misuse) and better protection while supporting (e.g., more networking by child protection with persons conducting practical support relating to hygiene and the household). This networking is required across agencies and sectors (welfare, health, education etc.). The use of identified networks to ensure collaboration with the necessary personnel to maximise PS-SP can broaden practices in relation to safety planning and safe practice as illustrated in Appendix A.

A safety dimension relates to the wider community, where members of public, such as the shop assistants, who might notice children doing shopping on their own, including for alcohol, should not just apply the law (not sell the alcohol) but also notify the local family resource center of their concerns for greater support to the family. From a developmental and/or management perspective, the PS-SP framework could inform critical thinking about the role of community in child protection and welfare. In the Roscommon case, for example, many referrals and expressions of concern came from the community but the report notes these were not generally taken seriously (McGregor and Dolan 2021, p. 145). However, it seems they may have been willing to play a more active part in the supportive protection if given the opportunity to do this. How to engage the public in the complex area of child protection and welfare, especially thinking about safety, confidentiality, and rights, is a challenging question. The PS-SP framework might offer more creative ways to think about this and explore (development) how the child welfare worker, for example, can mediate between their organisation, the law, and family privacy on one hand and the potential to harness wider family support and protection informally within communities on the other.

While acknowledging that risk assessment and tools to do this are central to child welfare and protection practice and many models prevail in different contexts to help move beyond defensive practice and reframe, more emphasis should be placed on its counterpoint, safety, in supervision discussions and analyses of practice. This contributes to existing models such as Signs of Safety in three ways: (a) the focus is on the practitioner and supervisor, (b) the focus is from micro to macro and chrono level capturing practitioners 'direct practice' and 'networked practice' across the eco system, and (c) PS-SP framework grounds the focus firmly on the duality of support and protection that, we contend, features across child welfare and protection systems irrespective of orientation and context. Why is this important? As referenced above, we know that wider inequalities have a major impact on individual family and child protection matters. Therefore, a wider frame for supervision is essential to mediate this and add to what is already known from evidence and experience. For example, we already know about the importance of a focus on safety in relation to feeling safe, keeping families safe, and safe supervision. Of note also, a wide range of studies conclude that the supervision process works best if it offers a space to explore emotions, develop knowledge and skills, and shape and/or improve social work practice decision-making. The use of supervision and other management techniques to enable practitioners to explore emotions is particularly neglected and lacking in practice, yet is highly relevant (Turney and Ruch 2018), as well as an organisational-level focus on safety in relation to health and safety policies and safety guidance and support for home visiting and day-to-day practices. We also have macro-level policy and legislation focused on safeguarding, but, again, the emphasis tends to be on the micro- and inter- personal level.

We have extensive literature on collaboration in child welfare and know its importance (Bruning and Doek 2021, p. 253; Churchill and Fawcett 2016, p. 313; Katz and Hetherington 2006, p. 433; Rácz 2015) and interagency cooperation (McGregor and Devaney 2020b, p. 283; Rácz and Bogács 2019, p. 156). It is also well established that partnership with service users is a key aspect of effective practice (Malone and Canavan 2021; Churchill and Fawcett 2016; Spratt et al. 2015; Connolly and Devaney 2018, p. 5; Meysen and Kelly 2018, p. 228). Yet, as Cortis et al. (2019, p. 56) warned, embedding relational approaches in child welfare practice remains difficult due to the systemic difficulties that Anglo-welfare states face in terms of escalating reports of maltreatment, increasing complexity of client circumstances, and rising resource pressures and costs. This again points to the need for a broader approach and more explicit framing in relation to safety, which underpins the intentions and complex context of much child welfare practice.

However, we argue that safety needs to be viewed from a wider dimension and we hope this discussion of the PS-SP framework in the context of safety in supervision adds to and complements existing approaches. It is also intended to offer ideas for how supervision models can be developed to reflect the complexity of child protection as evidenced from the ethnographic observations and experiences of those working directly with children and families, from the perspective of children and families themselves, and also from the immense research and data available. As illustrated in Table A1, safety comes into the frame across the eco system with regard to safe practice, safe communities, safety in the Person–Process context, safety across the eco-context, and so on. Building on what is already known and understood about safety, more explicit use of the PS-SP framework can more assertively inform safe practice taking into account the person (e.g., emotion, psychological issues and feelings), the process (safe balancing of protection and support), the context (detailing safety practices from micro to macro level), and time (explicitly exploring chrono level challenges of system change, managerialism, and resource management, for example).

Table A1 provides an example of how this could be applied to the case study if intervening in the present day. Reflecting on the Roscommon Case from this perspective this offers two important features. Firstly, an overarching systematic focus on PS-SP across all aspects of the support and protection work. This would lead to greater collaboration and networking between the different services and a commitment to approaching the case as a team, and not a single agency/practitioner who has sole responsibility at any point in time. Secondly, using PPCT as the tool to inform discussions ensures that practice and safety issues are not confirmed to micro- and mezo-level issues only but automatically are considered holistically from the ecological perspective. This means that application of models, such as Signs of Safety, could have a wider framework for reflection so that strengths and facilitators (for example) are not only focused on the child, family, and immediate context but also on the wider socio-economic and organisational context. All of this work in the present must be informed by a partnership and participative approach with children and families.

## 5. Conclusions

As mentioned, authors A and B put forward a model of PS-SP practice based in a broad ecological framework with an emphasis on networking. Recognising the need to 'test' this model in practice its value and significance is well debated. However, having reflected further on the model, we are now of the view one notable and necessary addition in the model is consideration of safety within this ecological context and with more specific attention to models and processes of supervision. This model encourages workers and supervisors to reframe the focus on risk and think also about its counterpoint, safety. We now contend that a safe milieu for the supervisee (and the supervisor) is required to ensure acceptance and understanding in their role to protect and support. We proposed some exploratory and reflective prompts to support supervisees and supervisors to think about safety in their role by reference to the Roscommon Case example and the illustration provided in Appendix A. Our argument is that it is practitioners and supervisors who are

best placed to populate and develop such a framework relating to their own practice. This then creates the opportunity to evaluate its usefulness for supervision with the intention of enhancing practitioner safety that, in turn, should lead to greater potential to engage directly with children and families, and indirectly through networking and collaboration, to achieve enhanced safety as a core and central outcome of any child protection and welfare system.

Supportive protection and protective support need to happen within the context of the ecological model and a networked approach that provides scope for a holistic and integrated response. It needs to be applied and tested by supervisors and supervisees to advance and develop this as it is in this ethnographic space of direct engagement in child welfare that the depth of mediation of protection and support is crystallised. We see those in practice, and those they are practising with as best placed to inform and expand the tremendous body of literature and research we have with regard to how best to mediate this space. A space we know is mostly a complex duality of PS-SP across many domains. Our argument is that while we will always need thresholds and levels of need/risk to inform decision making and practice, we also need to have an overarching reflective scaffold to see the whole context, and the wider set of circumstances in which children and families live. The PS-SP framework allows space for reflection, critical thinking, and a focus on safety without becoming subsumed into the more micro aspects of the 'case'.

This paper is written in the context of a general commitment to an increased pursuit of evidence-based policy and practice to ensure effective service provision across many jurisdictions (e.g., Spratt et al. 2015, p. 1522; Xu et al. 2018, p. 117). It reinforces the evidence from Lonne et al. (2021, p. 1), who identified the increased use of practice frameworks being used in conjunction with risk assessment tools aimed at promoting equality and quality service provision (Heino 2012, p. 158). In previous work, we offered an evaluation frame including a revised Hardiker model (Hardiker et al. 1991) which has been presented as a tool for diverting families with complex needs that do not fit either universal or protection intervention thresholds to an appropriate model of support (McGregor and Devaney 2020a, pp. 277, 287). In this paper, building on the PS-SP framework from (McGregor and Devaney 2020b), we hope to have added to this repertoire by focusing on another important dimension of child welfare practice, safety.

Reiterating an argument previously made (McGregor and Devaney 2020b), in order to truly test the framework, we need practitioners and supervisors to engage in its use in supervision and case work. As real-world experts, they are the ethnographers of practice, and are best placed—indeed, maybe the only ones placed—to articulate and ground the theorisation and conceptualisation of PS and SP which, are well established within the complex reality of child protection and welfare practice.

**Author Contributions:** All authors contributed equally. All authors have read and agreed to the published version of the manuscript.

**Funding:** This research received no external funding.

**Institutional Review Board Statement:** Not Applicable.

**Informed Consent Statement:** Not Applicable.

**Data Availability Statement:** Not Applicable.

**Conflicts of Interest:** The authors declare no conflict of interest.

## Appendix A

**Table A1.** Illustrative Example of Case Study as Applied to Child Protection and Welfare Social Work.

| | Safety (keeping children safe and safety of practitioners) | | | |
|---|---|---|---|---|
| | **Management** (competent, accountable performance or practice) | **Mediation** (engaging the individual with the organisation/s) | **Development** (continuing professional development) | **Personal Support** (supportive for the practitioner) |
| **Bio-eco level** | | | | |
| **Person** | Ensuring regular supervision using PS-SP framework including leadership on ensuring everyone knows CP is their business<br>Ensure clarity for each practitioner on their role and purpose<br>Clarity on expected outcomes and associated plan of action for each child<br>Clarity on process for risk escalation<br>Ensure each child's views are heard, recorded and presented in appropriate fora<br>Protected caseloads for new or inexperienced workers | Reflecting on skills, knowledge and values to ensure balance of duality of PS-SP<br>Clarity of role and of role of other colleagues and agencies | e.g., Training needs in:<br>Networking and build confidence to engage across support and protection services<br>Specialist training (e.g., in this case attachment, working with resistant parents,<br>recognizing risk, the impact of chronic neglect, participation and involvement of children and young people,<br>Use of standardized common assessment framework | Use current emphasis of safety in supervision<br>Space to talk about emotions, psycholgical impact, how to work with parents when they are resistant, being safe enough to be confident to network and call out protection needs<br>Be able to identify challenges and training needs |
| **Process** | Explicit use of PS-SP framework to discuss cases and apply to practice models | Using the PS-SP framework as guide for supervision and mediation actions<br>Duality in practice across sectors<br>Engaging individual with organization/mediation function<br>Mediation as capacity to act on behalf of organization and work with others—responsibility for CP across system (e.g., mediation between child welfare org, health org, home help, school and legal services)<br>Clarity of role and of role of other professionals and agencies | Developing practice with focus on direct practice and networked practice<br>Continuing Professional Development (CPD) required to support this<br>e.g., in this case, leadership, advanced skills in balancing support and protection, and training in specialist work with complex cases would have enabled a more integrated and potentially more supportive AND protective SAFE engagement with the family | Feeling safe to engage in networking in unsafe communities<br>Using this case example:<br>Engaging extended family, other professionals, and the community in the process of assessment, intervention and reviewing needs of children and dual protective/supportive focus |

**Table A1.** *Cont.*

| | | Safety (keeping children safe and safety of practitioners) | | | |
|---|---|---|---|---|---|
| | | **Management** (competent, accountable performance or practice) | **Mediation** (engaging the individual with the organisation/s) | **Development** (continuing professional development) | **Personal Support** (supportive for the practitioner) |
| **Bio-eco level** | | | | | |
| **Context** | **Micro** | Risk management discussed in context of safety outcomes Use concept of PS-SP to broaden the concept of safety planning | Mediation is framed more around risk/safety dichotomy within PS-SP context across each of the levels Mediation is applied across the system considering and inclusive of all stakeholders across all levels | CPD specialist training, e.g., in attachment | Safety discussions about micro-level practice—safe children, safe families, safe practitioners How can systems and practices ensure safety in networking? Safety discussions about exo-macro level indirect practice and networking—e.g., creating safety nets through family and social support |
| | **Meso** | Focus on relationship with supervisor? Is this a safe space? Workers in this case needed a safe environment to discuss their concerns etc. | | | |
| | **Exo** | What organisational management systems are in place for safety protocols—are they sufficient? Consider 'safety nets' 'safe communities' 'safety organised practice' | | Developing networking practices: interdisciplinary—targeted services, community organizations—reflections on 'safe collaboration' | |
| | **Macro** | Wider focus on CORU proficiencies—for e.g., on safe practice/wider regulatory concerns? Consider 'cultural safety' | | | |

**Table A1.** *Cont.*

| | | Safety (keeping children safe and safety of practitioners) | | | |
|---|---|---|---|---|---|
| | | **Management** (competent, accountable performance or practice) | **Mediation** (engaging the individual with the organisation/s) | **Development** (continuing professional development) | **Personal Support** (supportive for the practitioner) |
| **Bio-eco level** | | | | | |
| **Time** | **Chrono** | Recognise that management of CPW in this time is, by nature, complex balancing of PS-SP in context of managerialism etc. Recognize and address pressure of competing demands and resources | Mediation for safety taking into account children's and young person's participation and voice | Consider recent learning to inform implications of working with complex families With resistant parents. | Where are the 'safe spaces' for practice in your context/moment Applying to the case example: how can practitioners create a safe space with other practitioners to ensure no one practitioner feels solely responsible A shared sense of PS–PS |
| | **Moments** | Using case example instilling confidence in practitioners to base decision making on observations, concerns etc. which were noted repeatedly Using supervision to help practitioner to 'join' the experiences over time of home visits, reports received, ongoing high level of need and risk for the children involved, lack of positive progress or change Recognise emotion, feelings and impact on day-to-day practice actions Assurance and confidence to escalate risk management and make hard decisions regarding alternative care etc. | In depth analysis in specific case discussions of mediation of SP-PS to enhance safety | | Having space in supervision to reflect on the different levels of practice needed in order to ensure safety |

**Note**

[1] See also McGregor and Dolan (2021), for example of use of this case study to reflect on support and protection across the lifecourse.

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
