# Peer review of "Protective Support and Supportive Protection: Critical Reflections on Safe Practice and Safety in Supervision"

_socsci, doi:10.3390/socsci11070312_

Round 1

Reviewer 1 Report

Manuscript Revision

Protective Support and Supportive Protection: Critical Reflections on Safe Practice and Safety in Supervision

This manuscript aspires to contribute to the important field of social work supervision in child protection settings. Building on their prior work on the "protective support-supportive protection (PS-SP) model," the authors seek to expand the model and address the issue of supervision in CP practice. That is, they suggest that the model can contribute to the inclusion of contextual and structural dimensions in supervision. Although this is a much-needed addition to the field and despite the authors’ prior valuable work, I found the manuscript lacking and in need of major revision if it is to achieve its aims and be published. 

I believe the main shortcoming of the manuscript is a lack of connection to actual practice and to the ways in which the authors’ suggestions can be translated into supervision. Without these links, the manuscript currently reads like a long (albeit well written) literature review, and the appendices, which seem very relevant, serve as attachments to it.

Accordingly, I suggest the following major changes. First, the authors should add examples from practice/supervision in the form of either one in-depth case study or several examples. Some kind of substantial reference to the implementation of the model is essential. The appendices seem very practical, and it would be very helpful if the authors could better explain them and clarify how they can assist supervisors. In addition, a clearer (brief) outline of the PS-SP in the introduction would be helpful. Since these suggestions will require making a major addition to the manuscript, I suggest significantly shortening the second section—on child welfare orientations. This is only the background of the main focus of the manuscript, which is supervision.

I believe that making these changes will help this manuscript fulfill its potential.

Author Response

Draft manuscript

Protective Support and Supportive Protection: Critical reflections on Safe Practice and Safety in Supervision

Reviewer’s comments   

Authors response   

Reviewer 1  

I believe the main shortcoming of the manuscript is a lack of connection to actual practice and to the ways in which the authors’ suggestions can be translated into supervision. Without these links, the manuscript currently reads like a long (albeit well written) literature review, and the appendices, which seem very relevant, serve as attachments to it.  

Thank you for taking the time to read this draft manuscript and for your feedback. It is much appreciated.  

In response to this observation, we have revised the paper to ensure a strong connection to practice. Using a published case study as an illustrative example we have discussed the application of the PS-SP framework to practice and applied it to Appendix A.   

First, the authors should add examples from practice/supervision in the form of either one in-depth case study or several examples   

 Thank you for this suggestion.

An Irish case study based on a published Inquiry Report has been included in the discussion.  

Examples of applying the PS-SP framework and considering the issues raised in the paper based on the case study have been included. 

Some kind of substantial reference to the implementation of the model is essential. The appendices seem very practical, and it would be very helpful if the authors could better explain them and clarify how they can assist supervisors.  

Implementation of the framework based on a case study has now been included. The case study has also now been applied to Appendix A with a particular focus on using it to support the supervision process.   

In addition, a clearer (brief) outline of the PS-SP in the introduction would be helpful.   

A summary of PS and SP and its potential for use is now provided in the introduction.   

I suggest significantly shortening the second section—on child welfare orientations. This is only the background of the main focus of the manuscript, which is supervision.  

Thank you for this suggestion.

This section has been shortened considerably.  

Reviewer 2 Report

Many thanks for giving me the opportunity to review this paper. I really enjoyed reading it, and found the engagement with the literature on supervision particularly useful and extensive. Upon reading I felt that in order to be publishable this paper required some revision. These revisions were largely to assist the reader in understanding the approach being described. While I appreciate that the authors have described the approach in other papers, I did not feel that enough of an explanation was provided in order that this paper could stand on its own. As someone who was unaware of the approach being described, I found areas of the paper hard to follow and required further information in order to fully engage with the text. There were also some individual sentences that were hard to follow. I detail where improvements could be made below:

Section 2: Overall I felt this section was comprehensive and very well written. There were a handful of points (below) where I felt greater consideration of risk language and risk-focused systems were needed.

- Page 5, line 248, I didn't understand this sentence about family support, please check to see if a word is missing

- Page 6, line 271-272, the authors state that the underpinning aim of any child welfare system is safety. However, there is much literature to suggest that the underpinning aims are disputed and various - and for some the aim is risk management (without much consideration to safety at all). This tension is noted later in the paper but I felt this nuance needed to be recognised earlier in the paper and particularly here

Page 6 - line 275-278, which 'we' is being referred to and could a reference be offered to support the claim being made?

Page 6 - line 284 - 286: reference made to a focus on relational and psychological safety but not clear what the point is - focus on this as opposed to something else? 

Section 3: Good overview of supervision literature provided however it is here that I felt a more specific explanation of the approach needed to be included. I know the framework is in an appendix but arguable it, or a reduced version of it, would be better placed in the main text, with points made in respect of it. Without this section 3 reads as a summary of key thinking in respect of supervision but without clear account of the approach being suggested. As the authors then move straight into a discussion I found myself going back through the paper to more clearly find the framework being discussed. 

Section 4: 

Page 8, line 386-387, I did not understand the following sentence: 'The SP-PS framework serves to ground...'

Overall the discussion and conclusion were hard to follow without a clearer account in Section 3 of what the SP-PS framework was - and how the addition of 'safety' changed the previous framework. If the text on the framework was easily segregated from the wider discussion on supervision all references to the framework would have been easier to follow - as to would have been commentary on the opportunities and challenges of using it.

Author Response

Draft manuscript

Protective Support and Supportive Protection: Critical reflections on Safe Practice and Safety in Supervision

 Reviewer 2 Comments

  Authors Response

I suggest deleting references in the abstract and to review them in APA 7th ed. Revise the abstract to be more structured and become clearer.   

Thank you very much for taking the time to read our manuscript and for your feedback. It is very much appreciated.

References have been deleted from the abstract. The abstract has been revised to improve clarity, focus and structure.  

It is said that a model was developed by the authors, but little information is presented. Describe it briefly. The same about "Appendix C", if the reference is missing it is impossible to get deeper information. Add it or add the references 2020a and 2020b. Therefore, the manuscript seems to be very close to previous authors' publications, which undermines this article.   

A summary of the PS and SP framework and its potential for use is now provided in the introduction. 

Reference to Appendix C has been removed from the paper. We have focussed instead on applying the PS and SP framework to a published case study.  

Was the model that is presented an object of testing, and validation?  

Thank you for this question and prompting clarity on PS – SP  

We emphasise that PS-SP is a broad framework for critical analysis, reflection and thinking about the complex field of supporting and protecting children and families that can bolster practice and supervision discussions.  

The framework has not been tested; we see the way to test this framework as encouraging practitioners and students to use it to reflect on family's needs and circumstances.  This is our intention.

How was the process of writing the manuscript? The design, methodology, search, and so on, please present.   

This manuscript is based on a body of work built up over time by the authors. While the material presented draws from earlier articles it progresses in this paper to consider support and protection in child protection and welfare work with a focus on supervision and safety in practice.  

As noted, the framework has been applied to a case study, a current literature search was conducted on CPW and on supervision and safety in supervision.  

Reviewer 3 Report

Dear authors,

The topic presented is relevant to the child welfare/protection system, namely supervision in order to improve the quality of agencies' work.  

There are some comments below that could help strengthen the manuscript. 

I suggest deleting references in the abstract and to review them in APA 7th ed. Revise the abstract to be more structured and become clearer. 

It is said that a model was developed by the authors, but little information is presented. Describe it briefly. The same about "Appendix C", if the reference is missing it is impossible to get deeper information. Add it or add the references 2020a and 2020b. Therefore, the manuscript seems to be very close to previous authors' publications, which undermines this article. 

Was the model that is presented an object of testing, and validation?

How was the process of writing the manuscript? The design, methodology, search, and so on, please present. 

All the best.

Round 2

Reviewer 1 Report

Manuscript Revision – R1

Protective Support and Supportive Protection: Critical Reflections on Safe Practice and Safety in Supervision

The authors made substantial changes that improved the article. The manuscript is now much closer to actual practice and the potential of the framework for supervision is much clearer. I believe that the framework can make truly contribute to practice. Nevertheless, I am compelled to point out several shortcomings that require further work on the manuscript. I detail them here:

1. Section 2: Although you shortened it, the section is still very dense. I am not sure how your review of the literature contributes to your bottom line (if I have understood it correctly): "Overall, we can conclude from the literature that while there is a prevalence of ongoing dynamic change processes across various jurisdictions …. there is limited attention to the supervision and support for the practitioner to implement these practices and systems in a day-to-day basis." Moreover, I found it challenging to follow the ideas in this section. For example, you discuss the differences between orientations in terms of service provision (based on the example Kojan and Lonne). Then you discuss the ecological perspective and its challenges without relating to its relevance in the context of differing orientations. It seems that what you want to present are the challenges of current child protection practice across jurisdictions but you do not state this explicitly and it is not clear. I highly recommend clarifying the important points of this section and keeping it brief. I’m not sure if the many examples are beneficial in this case.

2. Section 4: The idea of using the case example is of course welcome, but in my opinion, it requires more work. First, the structure of the section is too loose and hard to follow. Second, whilst you describe the case in general, your actual discussion doesn’t really link to the case.  For example, you describe the critique regarding the lack of coordination between services and suggest that a PS-SP would include a shift to a more holistic framework and necessitate addressing the ecological context and so on. However, since you don’t describe what this would have looked like in this specific case, the case doesn’t contribute to the discussion. Another example: you mention that the engagement with the extended family wasn't sufficient and that a PS-SP would require a focus on creating a supportive environment via engagement with the family. Again, what would such a shift look like in this case?  You use the Roscommon case to point to the deficiencies of the system but not to portray how your alternative framework could have helped the actual social workers in the case at a practical level. This lack of connection is also reflected in Appendix A, which isn’t linked to the specific family in the case. If you want to retain the case study, you need to use it in a better way. Another option could be to walk readers through how each level of the model can contribute to supervision (perhaps with brief examples for each level).  I am aware that this is once again a substantial comment, but I feel it is essential for the article to be focused and impactful.

3. The text requires proofreading and editing.  In several instances, parentheses and apostrophes are opened but then not closed (e.g., line 28; lines 389–390; lines 410–412; line 215), full stops are missing or misplaced (e.g., line 553; line 414), double spaces appear at the beginnings of sentences throughout the manuscript, and some of the citations do not follow APA guidelines (e.g., line 252; line 296). In addition, a reference to Morrison is missing.